# Jointly Modeling Aspect Information and Ratings for Review Rating Prediction

Qingxi Peng [1], Lan You [2,*], Hao Feng [1], Wei Du [1], Kesong Zheng [1], Fuxi Zhu [1] and Xiaoya Xu [2]

1. School of Information Engineering, Wuhan College, Wuhan 430212, China
2. Faculty of Computer Science and Information Engineering, Hubei University, Wuhan 430062, China
* Correspondence: yoyo@hubu.edu.cn

**Abstract:** Although matrix model-based approaches to collaborative filtering (CF), such as latent factor models, achieve good accuracy in review rating prediction, they still face data sparsity problems. Many recent studies have exploited review text information to improve the performance of predictions. The review content that they use, however, is usually on the coarse-grained text level or sentence level. In this paper, we propose a joint model that incorporates review text information with matrix factorization for review rating prediction. First, we adopt an aspect extraction method and propose a simple and practical algorithm to represent the review by aspects and sentiments. Then, we propose two similarity measures: aspect-based user similarity and aspect-based product similarity. Finally, aspect-based user and product similarity measures are incorporated into a matrix factorization to build a joint model for rating prediction. To this end, our model can alleviate the data sparsity problem and obtain interpretability for the recommendation. We conducted experiments on two datasets. The experimental results demonstrate the effectiveness of the proposed model.

**Keywords:** rating prediction; matrix factorization; product review; aspect analysis





## 1. Introduction

E-commerce websites enable people to rate products and services with 1 to 5 stars after purchasing goods. These ratings are important to both merchants and customers. Merchants can use ratings to improve their production and sales strategies, while potential customers can use them to make better decisions. However, the volume of reviews is growing so rapidly that it is becoming increasingly difficult for users to browse reviews to find relevant information. Therefore, review rating predictions have become an extensively investigated issue in both academia and industry. The predictions enable researchers to estimate how satisfied a user will be with a product, without some extra text.

Most of the previous solutions consider the rating prediction as a recommendation system. The concept of context-aware recommendation technology proposed that context information can be introduced into the recommendation system, thereby improving the accuracy of the recommendation [1]. The use of contextual information to improve recommendations has experienced an upsurge in interest in the recommendation systems community. On E-commerce websites, users tend to write a review when they vote on products or services. Based on the above ideas, many works exploit various features from the review text, such as words, patterns, syntactic structure, and semantic topics, to improve the performance of rating prediction [2–8]. The above studies usually exploit text-level or sentence-level information in rating prediction. Although review analysis at the document level and sentence level is useful, it is still coarse-grained. It is worth noting that reviewers usually describe aspects of products or services to express their sentiments and convince other people. To obtain a more fine-grained review analysis, we need to delve into the aspect level. Aspect-level review text analysis is considered to be a fine-grained analysis in a large number of works [8–16]. Our focus of this paper is on how to use aspect-based

information in reviews to improve the accuracy of rating prediction. We first present a simple but effective rule-based algorithm to extract the aspect and corresponding sentiment from reviews. Then, we compute the aspect-based user and product similarity from the review text. Finally, we integrate the similarity into matrix factorization to obtain a joint model, thereby improving the accuracy of rating prediction.

In this paper, we propose a novel joint model, which incorporates the aspect-based product similarity and the aspect-based user similarity into matrix factorization. We first present a simple and effective aspect and corresponding sentiment extraction algorithm and then apply it to represent the review. Then, aspect-based products, as well as user similarity, are proposed. Finally, we propose a joint model, which incorporates the similarity measure into matrix factorization. Rather than performing context pre-filtering or post-filtering on the context recommendation [1], we model the aspect-based information in a single learning stage, which enables us to explore the implicit information of users and products simultaneously.

The main contributions of this study are summarized below:

1. We present simple and powerful aspects and corresponding sentiment extraction algorithms and apply them to represent the review text.
2. Two aspect-based similarity measures according to users and products are proposed.
3. We propose a joint model, which incorporates aspect-based information into matrix factorization for review rating prediction.

The remainder of the paper is organized as follows: Section 2 discusses related work on review rating prediction and matrix factorization techniques. In Section 3, an aspect and corresponding sentiment algorithm is proposed, which combines a bootstrap algorithm and sentiment lexicon. Reviews are represented by aspects and sentiment polarity. Then, aspect-based product and user similarity are proposed. Moreover, we propose a model to incorporate the above similarity measure into the matrix factorization algorithm for review rating prediction. Section 4 presents the empirical experiments used to evaluate the proposed model. Finally, the conclusions of our study are given in Section 5.

## 2. Related Work

### 2.1. Review Rating Prediction

Early researchers generally adopted classification or regression methods for rating prediction. In their study [17], Pang and Lee regarded rating prediction as a multi-classification problem and used classification and regression methods to find a solution. Goldberg and Zhu presented a graph-based semi-supervised learning algorithm to address the problem [18]. Lu et al. proposed an approach to predict the rating according to the strength of adverbs and adjectives in the review text [19]. Qu et al. introduced a bag-of-opinion to represent the review and adopted a constrained ridge regression algorithm to handle the rating prediction [20].

Many recent researchers have regarded rating prediction as a recommendation problem and exploited review text to help improve prediction performance. Wang et al. proposed a probabilistic rating regression model [8]. They also proposed a unified generative model for prediction, which did not require pre-specified aspect keywords [21]. In another study, Li et al. [22] modeled user, product, and text features as a three-dimension tensor to improve the performance of rating prediction. McAuley and Leskovec proposed a probabilistic model that combines latent rating dimensions with latent review topics for rating prediction [5]. Gao et al. modeled the rating as the similarity between user and product. They combined the topic modeling and regression model to predict the rating [23]. Tan and colleagues [24] exploited the topic-based user preference similarity in a traditional collaborative filtering algorithm to solve the data sparsity problem. Lei et al. proposed a matrix factorization method that incorporated three factors—user sentiment similarity, interpersonal sentimental influence, and product reputation similarity [7]. Yu et al. proposed a model combining the latent factor model and the latent Dirichlet Allocation [3]. By combining user sentiments in the review and the rating score, their model improved

the predictive ability. Yu et al. proposed a recommendation algorithm by integrating the user's social status with a matrix factorization model [25]. Ning et al. proposed a regression model based on generative convolutional neural networks [26]. They employed metadata instead of review text for rating prediction. Chambua et al. proposed a tensor factorization model with the semantic similarity between review texts [27]. Wu et al. proposed an enhanced review-based rating prediction by exploiting aside information and user influence [28]. Their model achieved 1.32% improvements on average in terms of MSE compared to existing models. A few existing studies employ attention mechanisms to differentiate the importance of reviews. Tay et al. proposed a multi-pointer learning scheme that learns to combine multiple views of user–item interactions [29]. Chen et al. introduced a novel attention mechanism to explore the usefulness of reviews, and proposed a neutral attention regression model with review-level explanations for recommendations [30]. Their model could not only predict precise ratings, but also learned the usefulness of each review simultaneously. Liu et al. proposed a hybrid neural recommendation model to learn the deep representations for users and items from both ratings and reviews [31]. Their model contains three major components, i.e., a rating-based encoder to learn deep and explicit features from the rating patterns of users and items, a review-based encoder to model users and items from text reviews, and the prediction module for recommendation according to the rating- and review-based representations of users and items.

The above works have improved rating prediction performance with the help of text-level review analysis. Aspect-level review analysis, however, can further improve the predictive ability.

### 2.2. Matrix Factorization Techniques

As the Netflix Prize competition has demonstrated, matrix factorization models are superior to classic near-neighbor techniques for producing recommendations [32]. Recommendation systems rely on different types of input data, often placed in a matrix, with one dimension representing users and the other dimension representing items of interest.

Matrix factorization models map both users and items to a joint latent factor space of dimensionality $f$, such that user–item interactions are modeled as inner productions in this space. Accordingly, each item $j$ is associated with a vector $V_j \in R^f$, and each user $u$ is associated with a vector $U_u \in R^f$. For a given item j, the elements of $V_j$ measure the extent to which the item possesses those factors, positive or negative. For a given user $u$, the elements of $U_u$ measure the extent of interest that the user has in items that are high on the corresponding factors, again, positive or negative. The resultant dot product captures the interaction between the user and the item's characteristics. This approximates user $u's$ rating of item j, which is denoted by $R_{uj}$, leading to the estimate

$$\hat{R}_{uj} = U_u^T V_j \tag{1}$$

For each given training case, the system predicts $R_{uj}$ and computes the associated prediction error.

The goal of rating prediction is, given training data $R_{uj}$, to find a mapping $U_u$ and $V_j$, such that

$$E_{uj} = R_{uj} - \hat{R}_{nj} \tag{2}$$

is a minimum, where $\hat{R}_{nj}$ is the predicted rating given as the product $j$ by the user $u$.

To learn the factor vectors ($U_u$ and $V_j$), the system minimizes the regularized squared error on the set of known ratings:

$$\sum_{u=1}^{K} \sum_{j=1}^{N} (R_{uj} - U_u^T V_j)^2 + \lambda_U || U ||_F^2 + \lambda_V || V ||_F^2 \tag{3}$$

Here, $R_{uj}$ is the training set, and $U_u^T V_j$ is the true rating.

The algorithm learns the model by fitting the previously observed ratings. However, the goal is to generalize those previous ratings in a way that predicts future, unknown

ratings. Thus, the system should avoid overfitting the observed data by regularizing the learned parameters whose magnitudes are penalized. The constant $\lambda_U$ and $\lambda_V$ control the extent of regularization and are usually determined by cross-validation.

Two approaches to minimize Equation (3) are stochastic gradient descent (SGD) and alternating least squares (ALS). We adopt SGD in this paper. To optimize Equation (3), the algorithm iterates over each rating on the training set. We set $\lambda_U = \lambda_V = \lambda$ for simplification. For each pair of $(U_u, V_j, R_{uj})$, the algorithm defines a new loss function $E_{uj}$.

$$E_{uj} = \frac{1}{2} \sum_{u=1}^{K} \sum_{j=1}^{N} (R_{uj} - U_u^T V_j)^2 + \frac{\lambda}{2} (\| U \|_F^2 + \| V \|_F^2) \tag{4}$$

For parameter $U_u$ and $V_j$, the SGD method first separately finds their partial derivatives.

$$\frac{\partial E_{uj}'}{\partial U_u} = E_{uj} \cdot (-V_j) + \lambda U_u \tag{5}$$

$$\frac{\partial E_{uj}'}{\partial V_j} = E_{uj} \cdot (-U_u) + \lambda V_j \tag{6}$$

According to the SGD method, the iteration equation is

$$V_j \leftarrow V_j + \gamma \cdot (E_{uj} \cdot U_u - \lambda \cdot V_j) \tag{7}$$

$$U_u \leftarrow U_u + \gamma \cdot (E_{uj} \cdot V_j - \lambda \cdot U_u) \tag{8}$$

The parameter $\gamma$ is the learning rate. Finally, we use submatrices $U$ and $V$ for rating prediction.

The other method of solution of the matrix factorization is the ALS method. The ALS techniques rotate between fixing the values of $U_u$ and $V_j$. When all values of $U_u$ are fixed, the system recomputes $V_j$ by solving the least-squares problem, and vice versa. This ensures that each step decreases the value of Equation (4) until convergence is achieved.

Many researchers have used the matrix factorization technique to solve the problem of rating prediction. Pero and Horvath incorporated ratings provided by users and opinions inferred from their reviews in matrix factorization [33]. Zhang et al. proposed a kernel-based attribute-aware matrix factorization model to integrate the attribute information of items into matrix factorization for rating prediction [34]. Zhang et al. proposed a framework that combined network embedding and probabilistic matrix factorization for improved predictive ability [35]. In this paper, we also take the above strategy, and integrate the fine-grained aspect-based information into a standard matrix factorization technique for rating prediction.

## 3. Methodology

Traditional methods either discard review text or treat all of the text as a whole. To combine the aspect information, we should resort to aspect analysis, including aspect extraction and aspect summarization. To make the method easier, we transform the aspect information into user similarity and product similarity. Then, we model the transformed aspect information in a classical matrix factorization model. Our model is concise and easy to interpret. First, we represent the review text with aspect information. Then, we compute the aspect-based product similarity and the aspect-based user similarity. Next, the aspect-based similarities are modeled in matrix factorization, to predict the review rating. Figure 1 shows the review rating prediction flowchart. The hyphen in the left figure represents the value that needs to be predicted, while the red number in the right figure represents the predicted value.

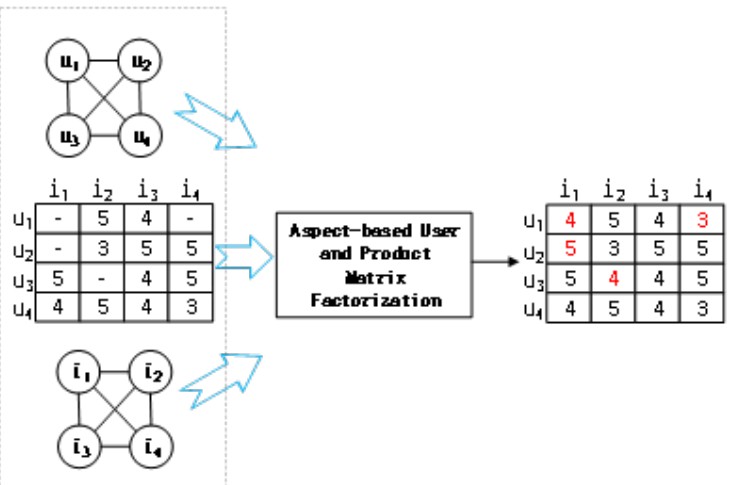

**Figure 1.** Review rating prediction flowchart.

In Table 1, we list the notations of the following parts.

**Table 1.** Notations.

| Symbol | Description |
| --- | --- |
| $U$ | $m \times f$ matrix, represents user's preference for a product |
| $V$ | $n \times f$ matrix, indicates that a product belongs to a preference |
| $R$ | rating matrix |
| $U_u$ | $f$ dimensional column vector of user $u$ |
| $V_j$ | $f$ dimensional column vector of product $j$ |
| $R_{uj}$ | rating of user $u$ to product $j$ |
| $\lambda$ | normalization parameter |
| $M_{jk}$ | product–aspect matrix, product $j$ to aspect $k$ , the value is +1, −1 or 0 |
| $S_{jn}$ | aspect-based similarity matrix between product $j$ and $n$ |
| $N_{uk}$ | user–aspect matrix, user $u$ to aspect $k$, the value is +1, −1 or 0 |
| $T_{um}$ | aspect-based similarity matrix between user $u$ and $m$ |
| $\alpha$ | weight parameter to balance product weight |
| $\beta$ | weight parameter to balance user weight |

*3.1. Aspect Sentiment Representation*

To exploit the aspect information of reviews, we should extract it from the review text. We adopt an aspect segmentation algorithm, presented in [8]. Given a collection of reviews and a set of aspect keywords, the algorithm splits the reviews into sentences with aspect assignments. We modified the algorithm with sentiment lexicons to separate the aspect and corresponding sentiments. Suppose a review $r$ = {<w1, q1>, <w2, q2>, ...< wN, qN>}, where $w_i$ is the aspect keyword, $w_k$ is the aspect category. $w_i \in W_k$, $q_i$ is the sentiment keyword, $\varrho^+$ represents a positive sentiment lexicon, and $\varrho^-$ represents a negative sentiment lexicon. $q_i \in \varrho^+ \cup \varrho^-$.

In Figure 2, user $u_1$ expresses four aspects concerning product $i_2$: breakfast, people, location, and room. With algorithm RAS (Algorithm 1), we can easily segment the review text into four aspects and corresponding sentiment polarities. We formally transform the review text as (breakfast, great), (people, nice), (location, good), (room, disappointed). It is important to note that the aspect keyword in the first aspect is breakfast. In our method, breakfast is included into the category of food. The aspect keyword in the second—people— is included into the category of staff. Since the positive sentiment value is set to 1, and the negative sentiment value is set to 0, we obtain (food, 1), (staff, 1), (location, 1), and (room, 0). It is important to note that our approach will not handle neutral words. Therefore, there are only two situations, positive and negative, in our method.

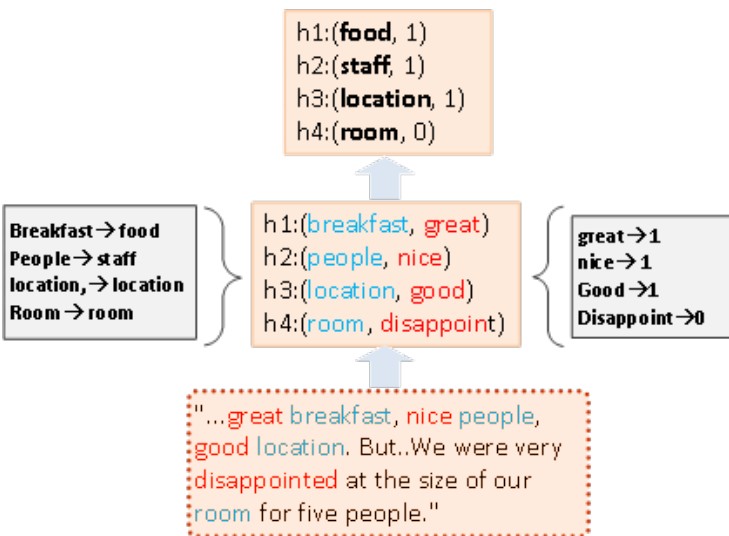

**Figure 2.** Review aspect and sentiment representation.

---

**Algorithm 1** Review Aspect and Sentiment Algorithm, RAS

---

1: **Input:** r = {<wi, qi> | i = 1, ..N} // review collection
2:         r = {w1, q2, ..wk} // aspect category
3:         D = $\varrho^+ \cup \varrho^-$ //*sentimentlexicon*
4: **Output:** r = {<wi, plt> | i = 1, ..N} // *plt* is Boolean value, 1 means positive sentiment, 0 means negative sentiment.
5: **For** i = 1 to **do**
6:     **if** $w_i \in W_k$**do**
7:         **if** $q_i \in \varrho^+$**then** $r = r \cup < Wk, 1 >$
8:         **if** $q_i \in \varrho^-$**then** $r = r \cup < Wk, 0 >$
9:     **End do**
10: **End do**
11: **Return r**

---

### 3.2. Aspect-Based Similarity Measure

3.2.1. Aspect-Based Product Similarity

In the above section, one piece of the review was represented by aspects and corresponding sentiments. In this section, we build a product–aspect matrix. We define the value of the product $j$ $s^{th}$ review aspect $k$ as $w_{sjk}$. One product may correspond to multiple reviews. For total S reviews, if there are more positive sentiments than negative sentiments in aspect $k$, we define $w_{sjk}$ as 1. If there are more negative sentiments than positive sentiments, we define $w_{sjk}$ as $-1$. Otherwise, we define $w_{sjk}$ as 0. Then, we obtain the product–aspect matrix M as

$$M_{jk} = \begin{cases} 1, & if \ \sum_{s=1}^{S} w_{sjk} > S/2 \\ -1, & if \ \sum_{s=1}^{S} w_{sjk} < S/2 \\ 0, & else \end{cases} \tag{9}$$

According to traditional item-based collaborative filtering, product similarity can be defined as cosine similarity. Here, we define aspect-based product similarity using improved cosine similarity [36]. The similarity matrix between product j and product n can be defined as

$$S_{jn} = \frac{\sum_{k=1}^{E}(M_{jk} - \overline{M}_k)(M_{nk} - \overline{M}_k)}{\left(\sqrt{\sum_{k=1}^{E}(M_{jk} - \overline{M}_k)^2}\right)\left(\sqrt{\sum_{k=1}^{E}(M_{nk} - \overline{M}_k)^2}\right)} \tag{10}$$

where $M_{jk}$ and $M_{nk}$ represent values in the aspects of products $j$ and $n$; $\overline{M}_k$ represents the mean value of aspect $k$.

### 3.2.2. Aspect-Based User Similarity

In this subsection, we first build a user–aspect matrix. Since one user may post multiple reviews, we choose the majority principle. We define the value of user $u$ $s_{th}$ review aspect $k$ as $w_{suk}$. One user may post multiple reviews. For the total reviews, if there are more positive sentiments than negative sentiments in aspect $k$, we define $w_{suk}$ as 1. If there are more negative sentiments than positive sentiments, we define $w_{suk}$ as $-1$. Otherwise, we define $w_{suk}$ as 0. Then, we obtain the user–aspect matrix $N$ as

$$N_{uk} = \begin{cases} 1, & if \sum_{s=1}^{S} W_{suk} > S/2 \\ -1, & if \sum_{s=1}^{S} W_{suk} < S/2 \\ 0, & else \end{cases} \tag{11}$$

According to traditional item-based collaborative filtering, user similarity can be defined as cosine similarity. We define aspect-based user similarity using improved cosine similarity [32]. The similarity matrix between user u and user n can be defined as

$$T_{um} = \frac{\sum_{k=1}^{E}(N_{uk} - \overline{N}_k)(N_{mk} - \overline{N}_k)}{\left(\sqrt{\sum_{k=1}^{E}(N_{uk} - \overline{N}_k)^2}\right)\left(\sqrt{\sum_{k=1}^{E}(N_{mk} - \overline{N}_k)^2}\right)} \tag{12}$$

where $N_{uk}$ and $N_{mk}$ represent values in the aspect of user $u$ and $m$; $\overline{N}_k$ represents the mean value of aspect $k$.

### 3.3. Joint Model for Rating Prediction

Standard matrix factorization can be expressed as Equation (4). Given user–product rating matrix R, it represents a rating from user $u$ to product $j$. $U_u^T V_j$ is the prediction rating. $R_{uj}$ is the actual rating. Now, the aspect-based user similarity and the product similarity are incorporated into the above objective function. Our joint aspect-based similarity model is as in Equation (13).

$$L(U, V) = \begin{cases} \frac{1}{2}\sum_{u=1}^{M}\sum_{j=1}^{N}(R_{uj} - U_u^T V_j)^2 + \frac{\alpha}{2}\sum_{j=1}^{N}\sum_{n=1}^{N}(S_{jn} - V_j^T V_n)^2 + \\ \frac{\beta}{2}\sum_{u=1}^{M}\sum_{m=1}^{M}(T_{um} - U_u^T U_m)^2 + \frac{\lambda}{2}(\| U \|_F^2 + \| V \|_F^2) \end{cases} \tag{13}$$

On the one hand, this model can alleviate the problem of low accuracy caused by sparse data. On the other hand, our model takes advantage of aspect information, which is less difficult than modeling review text directly.

The objective function is minimized by the SGD algorithm as Equations (14) and (15).

$$\frac{\partial L}{\partial U_u} = \sum_{j=1}^{N}(R_{uj} - U_u^T V_j)(-V_j) + \beta \sum_{m=1}^{M}(T_{um} - U_u^T U_m)(-U_m) + \lambda U_u \tag{14}$$

$$\frac{\partial L}{\partial V_j} = \sum_{u=1}^{M}(R_{uj} - U_u^T V_j)(-U_u) + \alpha \sum_{n=1}^{N}(S_{jn} - V_j^T V_n)(-V_n) + \lambda V_j \tag{15}$$

The stochastic gradient descent algorithm is as Algorithm 2. We call our method as Aspect-Based User and Product Matrix Factorization (AUPMF).

---

**Algorithm 2** Aspect-Based User and Product Matrix Factorization, AUPMF

---

1:  **Input:** $R$ // rating matrix
2:       $S_{jn}$ // aspect-based product similarity
3:       $T_{um}$ // aspect-based user similarity
4:       $\alpha, \beta$ // weight parameter
5:       $\lambda$ // normalization parameter
6:       $iter_{max}$ // iteration limit
7:       $\epsilon$ // stop condition
8:  **Output:** $\hat{R}$ // user–product rating matrix
9:  // data preprocessing
10: Initialize $U^{(0)} V^{(0)}$ with random value
11: t = 0; //Iteration number
12: $\tau = 0$; //Convergence flag
13: **Compute** $L^{(t)}$;       //Equation (13)
14: **While(** $t < iter^{max}$ **and** $\tau = 0$ )**do**
15:      $\eta = 1$;
16:      **Compute** $\frac{\partial L}{\partial U^{(t)}}, \frac{\partial L}{\partial V^{(t)}}$; //Equations (14) and (15)
17:      **While(** $L(U^{(t)} - \eta \frac{\partial L}{\partial U^{(t)}}, V^{(t)} - \eta \frac{\partial L}{\partial V^{(t)}}) \geq L^{(t)}$ )**do**
18:           $\eta = \eta / 2$;
19:           $U^{(t+1)} = U^{(t)} - \eta \frac{\partial L}{\partial U^{(t)}}, V^{(t+1)} = V^{(t)} - \eta \frac{\partial L}{\partial V^{(t)}}$; //Update
20:           **Compute** $L^{(t+1)}$;       //Equation (13)
21:           **If** $(L^{(t)} - L^{(t+1)} \leq \varepsilon)$
22:                $\tau = 1$;
23:      $t = t + 1$;
24:      **End**
25: **End**
26: **Return** $\hat{R} = U^{(t)T} V^{(t)}$

---

## 4. Experiments and Analysis

To evaluate the effectiveness of the proposed model, this section uses real-life review data to conduct experiments. First, we analyze the impact of the weight parameters on the proposed model. Second, we compare our proposed approach with five existing models to demonstrate our model. The third experiment studies the impact that matrix density has on the predictive ability of the model. The fourth experiment investigates the influence of the latent dimension. The experimental results demonstrate the effectiveness of the proposed approach.

In this section, we first describe the review data that we used for evaluating the proposed model and then discuss the experiments.

### 4.1. The Dataset and Preprocessing

Our hardware and software configuration are Intel(R) Core(TM)i7-5600U CPU with 2.60 GHz and 8.0 G memory, Windows 2012, Python 3.5.2, NLTK 3.0, Numpy 1.11.2, SciPy 0.17.0, Scikit-Learn 0.19.1.

### 4.2. Experimental Setup

Our experimental data come from the review data of Yelp. Yelp is a famous rating website that has large numbers of restaurants, shopping malls, hotels, etc. Yelp allows users to post review text and ratings on the website. After a series of preprocessing steps, we obtain the Yelp data as follows.

As shown in Table 2, our Yelp dataset includes two subsets: a restaurant dataset with 1,344,405 reviews, and a hotel dataset with 96,384 reviews.

**Table 2.** Yelp data statistics.

|          | Restaurant | Hotel   |
|----------|------------|---------|
| reviews  | 1,344,405  | 126,384 |
| products | 7438       | 2372    |
| users    | 19,150     | 12,305  |

We manually set the aspect seed keywords for restaurants and hotels as listed in Tables 3 and 4.

**Table 3.** Aspect seed words for restaurant dataset.

| Aspect   | Seed Words                                  |
|----------|---------------------------------------------|
| value    | money, price, dollars, cash, check, quality |
| service  | waiter, manager, staff, hostess             |
| meat     | beef, bbq, pork, hamburger, hotdog          |
| decor    | design, ceiling, decor, window, space       |
| dessert  | dessert, chocolate, ice cream, macaroons    |
| ambiance | ambiance, atmosphere, experience            |

**Table 4.** Aspect seed words for hotel dataset.

| Aspect              | Seed Words                       |
|---------------------|----------------------------------|
| room                | room, suite, view, bed           |
| value               | value, price, quality, worth     |
| location            | location, traffic, car, restaurant |
| cleanliness         | clean, dirty, maintain, smell    |
| check in/front desk | stuff, check, help, reservation  |
| service             | service, food, breakfast, buffet |

We conduct 5-fold cross-validation in the experiments. The data have been split into five parts. Four parts have been treated as training data, while the last part has been treated as the test data. This paper chooses the Mean Squared Error (MSE) as the evaluation standard. The MSE is defined as

$$MSE = \frac{1}{M} \sum_{u,j} (\hat{R}_{uj} - R_{uj})^2 \tag{16}$$

where $M$ is the total number of reviews in our collection. $\hat{R}_{uj}$ and $\hat{R}_{uj}$ are the predicted rating and the actual rating in the test data. The result is the mean value of five experiments. The metric measures how much our predicted rating deviates from the true rating. A smaller MSE value indicates better performance.

### 4.3. Baselines

We use several baselines to compare with our approaches.

1.  Basic Matrix Factorization **(BasicMF)**: Koren etc. propose the standard matrix factorization [32], which only uses rating to train the model. The BasicMF model optimizes Equation (4) using the SGD algorithm with Equations (7) and (8) until the iteration ends.

2.  Word-Based Similarity Matrix Factorization **(WSMF)**: WSMF directly uses word similarity in the review, to improve the standard matrix factorization. First, it lists all the words in the review text, and exploits TF-IDF to sort important words to build features. It then transforms the review into an N-dimensional vector. The similarity of reviews is the cosine similarity of the above two vectors. Finally, we incorporate the similarity into matrix factorization.

3.  Matrix Factorization with Bias **(BiasMF)**: BiasMF exploits user and product bias information in matrix factorization to improve the rating prediction [32].

4.  Sentiment-Based Rating Prediction method **(RPS)**: Lei etc. propose a sentient-based method [7]. It first builds a sentiment lexicon, and then calculates the sentiment of the review with a series of rules. Next, it proposes three important factors (user sentiment similarity, item reputation similarity, and interpersonal sentiment influence), and fuses them into matrix factorization.

5.  Hidden Factors and Hidden Item Topics **(HFT)**: The HFT model uses a traditional latent factor model to combine latent rating dimensions with latent review topics [5]. The accuracy of HFT is higher than that of the traditional LFM model. The HFT is a state-of-the-art algorithm for rating prediction.

### 4.4. Evaluation Results

To verify the effectiveness of the AUPMF model, we perform comparisons with existing models. We also employ five-fold cross-validation. All the results are represented by means and variance of five results.

#### 4.4.1. Impact of Weight Parameter

The weight parameters $\alpha$ and $\beta$, respectively, represent the proportion of aspect-based product similarity and user similarity in the proposed model. For the weight parameter $\alpha$, a larger $\alpha$ means that the joint model relies more on product similarity. On the contrary, a smaller $\alpha$ means that the joint model relies less on product similarity. If $\alpha = 0$, the joint model will not rely on product similarity, and it will only rely on user similarity to learn the latent factor vector. For weight parameter $\beta$, a larger $\beta$ means that the joint model relies more on user similarity. On the contrary, a smaller $\beta$ means that the joint model relies less on user similarity. If $\beta = 0$, the joint model will not rely on user similarity, and only relies on product similarity to learn the latent factor vector.

We use the AUPMF algorithm to conduct experiments on the restaurant and hotel datasets. First, we set the aspect number as 6. The aspect seed words are set manually as listed in Tables 3 and 4. To study the weight parameter $\alpha$ and $\beta$, we set the values of $\alpha$ and $\beta$ from 0 to 100, respectively, in steps of 10. We also set normalization parameter $\lambda = 1$, the number of latent features $f = 20$, and number of iterations to 1000.

Figure 3 shows how the weight parameters $\alpha$ and $\beta$ impact the rating prediction in the restaurant dataset. The weight parameters $\alpha$ and $\beta$ indeed influence the effectiveness of the proposed model. As $\alpha$ increases, MSE passes through a minimum, which means that the rating prediction initially goes up and then decreases. Parameter $\beta$ also has the same effect on rating prediction. It is shown in Figure 3 that in the restaurant dataset, the MSE has a minimum value of 1.312 when $\alpha = 20$ and $\beta = 60$. Therefore, the optimal values are $\alpha = 20$ and $\beta = 60$.

We continue the experiment with the above settings in the hotel dataset, and keep other parameters unchanged. Then, we study the impact that the weight parameters $\alpha$ and $\beta$ have on the rating prediction in the hotel dataset.

Figure 4 shows how the weight parameters $\alpha$ and $\beta$ impact the effectiveness of the rating prediction in the hotel dataset. The parameters $\alpha$ and $\beta$ indeed affect the performance of the prediction model. As above, the MSE shows similar trends for the restaurant and hotel datasets. With an increase in parameters $\alpha$ and $\beta$, the MSE value first decreases, which means higher accuracy of prediction. When a certain value is reached, the MSE increases with increasing $\alpha$ and $\beta$, which means lower accuracy.

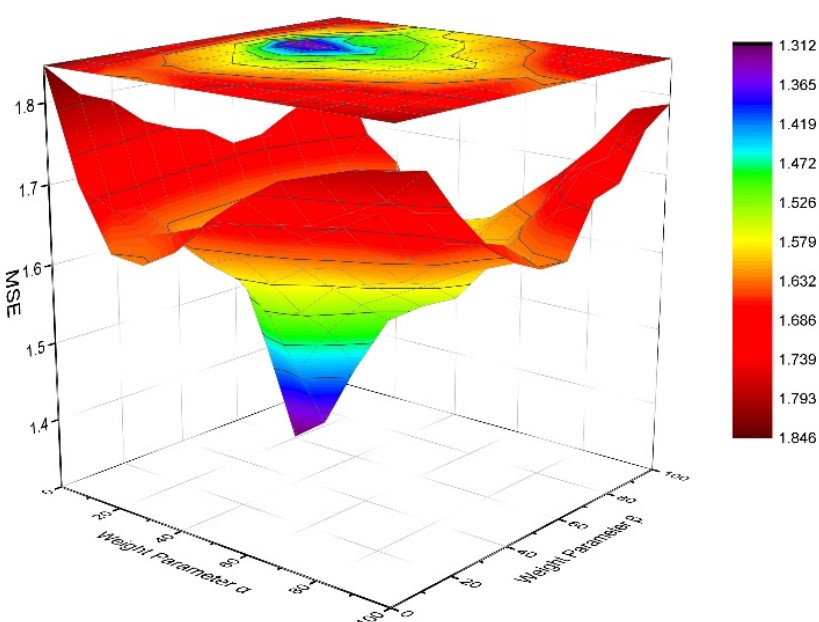

**Figure 3.** MSE vs. weight parameter in restaurant dataset.

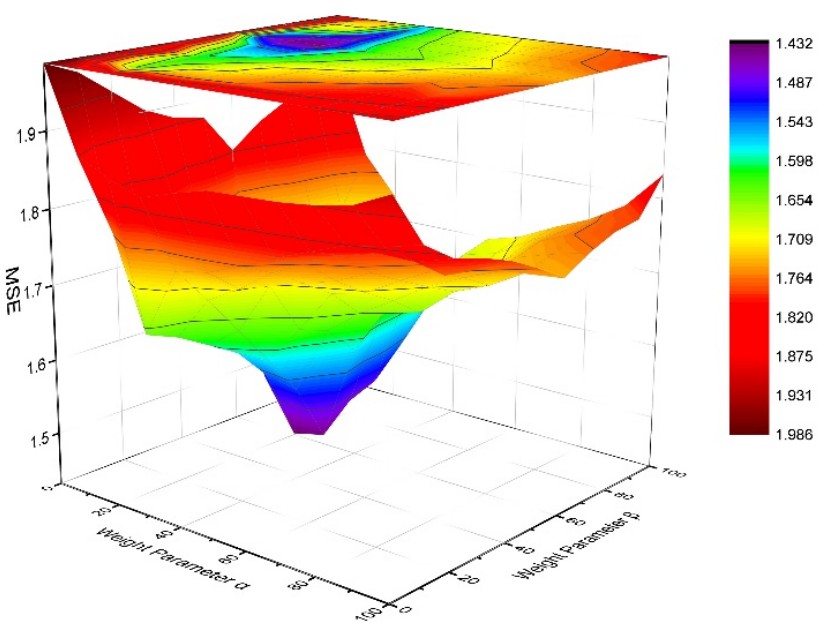

**Figure 4.** MSE vs. weight parameter in hotel dataset.

The data presented in Figure 4 also show that the optimal values of the weight parameters $\alpha$ and $\beta$ are the same in the two datasets. For the restaurant dataset, the model acquires the highest accuracy when $\alpha = 20$ and $\beta = 60$, whilst, for the hotel dataset, the model also acquires the highest accuracy when $\alpha = 20$ and $\beta = 60$. It can be seen from the experimental results that for the weight parameters $\alpha$ and $\beta$, $\beta$ is much larger than $\alpha$. It suggests that our model relies more on aspect-based user similarity than on aspect-based product similarity. In the review dataset, the products represent restaurant and hotels, and the number is relatively small compared with the number of users. Therefore, the impact of the product on the rating prediction is relatively small.

The value of MSE in Figure 3 is less than the value of MSE in Figure 4, indicating that our model's predictive ability varies for different datasets. The reasons for this will be discussed in Section 4.4.4. Now, we set $\alpha = 20$ and $\beta = 60$ in the following experiments.

### 4.4.2. Influence of Latent Dimension

For matrix factorization-based models, the latent dimension is an important parameter to tune. Our model involves such a parameter, $f$. In Section 4.4.1, we temporarily set it as 20. This section records how the number of latent factors influences the predictive ability of AUPMF. We vary it from 5 to 50 with a step of 5, and examine how the performance changes with regard to the latent dimension. As shown in Figure 5, using $f$ = 20 yields the best performance in the restaurant dataset, and $f$ = 25 in the hotel dataset. In order to facilitate the procedures, we still set $f$ = 20 in the following experiment.

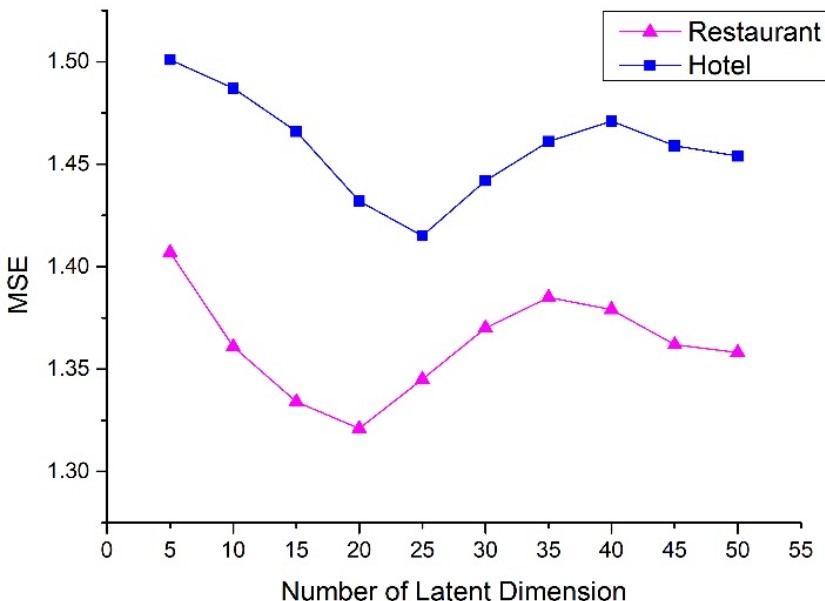

**Figure 5.** MSE vs. number of latent dimensions.

### 4.4.3. Comparison of Rating Prediction

Figure 6 compares the MSE for the two datasets determined using the six different models. The following conclusions can be obtained.

1.  Due to data sparsity, the standard matrix factorization could not achieve better results. The MSEs of BasicMF for the restaurant and hotel datasets are 1.740 and 1.762, respectively.
2.  WSMF performs worse than BasicMF, with MSEs of 1.920 and 1.971 for the two datasets, respectively. WSMF directly employs a word vector on standard matrix factorization, which reduces the predictive ability of the model.
3.  BiasMF employs the bias information of the user and product to improve the matrix factorization, gaining stronger prediction. The MSEs of BiasMF are 1.575 and 1.621, respectively, for the two datasets.
4.  The RPS model fuses several types of information to reduce the MSE; values of 1.437 and 1.534 were achieved for the two datasets, respectively. This model, however, relies on the sentiment lexicon, which affects the stability of prediction. It can be seen from Figure 5 that the deviation of several experiments is large.
5.  Compared with the above baseline models, the HFT's predictive ability is relatively strong. The average MSE was 1.346 and 1.458 for the restaurant and hotel datasets, respectively.
6.  The results obtained from the experiments indicate that AUPMF performs consistently and significantly better than the baseline methods. This is illustrated in Figure 6 in terms of the MSE. The average MSE of AUPMF is 0.03 lower than that of HFT for the restaurant dataset, which means higher accuracy. For the hotel dataset, the average

MSE of AUPMF is 0.03 lower than that of HFT, meaning that the predictive ability is stronger than that achieved using HFT.

All of the methods provide better prediction for the restaurant dataset compared with the hotel dataset. The reason is that the two datasets are different in sparsity. The hotel dataset is sparser than the restaurant dataset, and therefore the rating prediction is worse. We will discuss the effect of matrix density on rating prediction in Section 4.4.4.

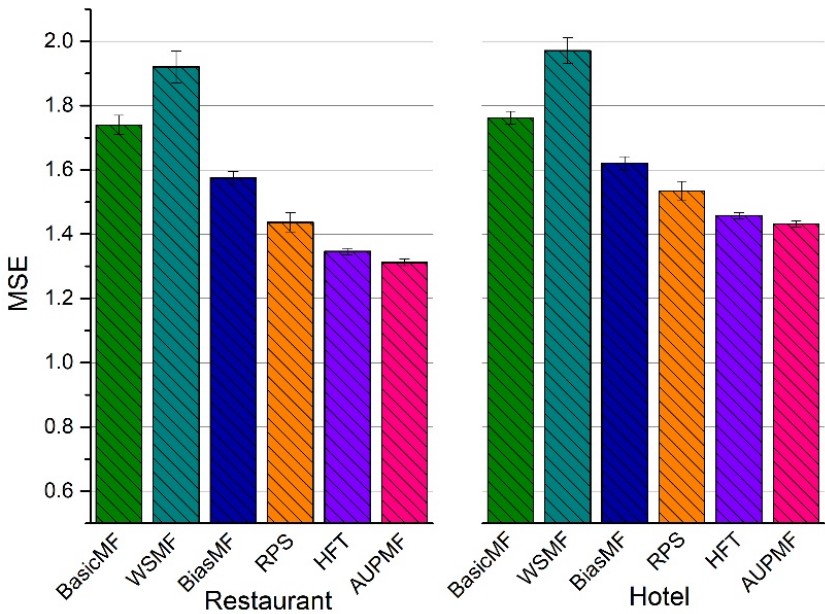

**Figure 6.** MSE vs. different models for two datasets.

### 4.4.4. Influence of Matrix Density

An important problem to be solved in this paper is the influence of matrix density on prediction. In order to examine the influence of matrix density, we conducted experiments with a range of different matrix densities. Suppose that m users post $x$ reviews on n products. The user–product rating matrix density is $\frac{x}{m \times n}$. To obtain matrices with different densities, we construct them from the original matrix according to the method proposed by Li et al. [22]. We only conduct this experiment in the restaurant dataset. We define a threshold value $\delta$. A large $\delta$ value means that users and products with large numbers of reviews will be kept, resulting in a dense matrix. A small $\delta$ value means that users and products with fewer reviews will be kept, resulting in a sparse matrix. There are four sub-datasets arranged on the X-axis of Figure 6. Their matrix densities $\delta$ are 0.034, 0.142, 0.207, and 0.292. The predicted MSEs for the different sub-datasets are plotted on the Y-axis. The experiments in this section are only carried out using the three best rating prediction models, i.e., RPS, HFT, and AUPMF.

As shown in Figure 6, the predictive ability of all models increases as the matrix density increases. When the matrix becomes dense, all models obtain more information and the performance of the different models will be improved.

As can be seen from Figure 7, both AUPMF and HFT outperform RPS in all sub-datasets. When the matrix density is 0.034, the MSE value of AUPMF is 0.03 and 0.13 lower than that of HFT and RPS, respectively. When the matrix density is 0.142, the MSE value of AUPMF is 0.03 and 0.14 lower than that of HFT and RPS, respectively. In one instance, HFT provides the best prediction; when the matrix density is 0.207, the MSE value of HFT is 0.01 and 0.06 lower than AUPMF and RPS. When the matrix density is 0.292, the MSE value of AUPMF is 0.02 and 0.07 lower than HFT and RPS. From the above analysis, we can see that the performance of AUPMF and HFT is relatively close. On three of four sub-datasets, the performance of AUPMF is better than that of HFT. Only when the matrix

density of the sub-dataset is 0.207, the MSE of AUPMF is higher than that of HFT, showing that its predictive ability is slightly lower than that of HFT. When the matrix density of the sub-datasets is relatively small (0.142 and 0.034), the performance of AUPMF exceeds that of HFT, showing the strong robustness of the AUPMF model.

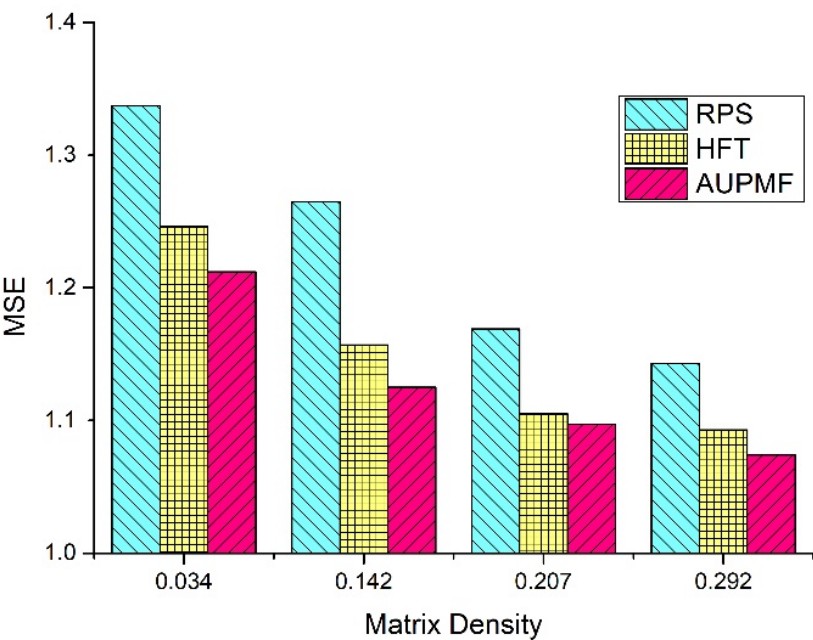

**Figure 7.** MSE vs. different matrix densities for restaurant dataset.

According to Table 2, the matrix density of the restaurant dataset is 0.0094, and the matrix density of the hotel dataset is 0.0043. In Figure 7, the matrix density of the constructed sub-datasets is higher than that of the original datasets, so the predictive ability of the model is improved. This also explains why the model proposed in Section 4.4.3 has stronger predictive ability for the restaurant dataset compared with the hotel dataset.

## 5. Conclusions

In this paper, we propose a joint aspect-based user and product model for review rating prediction. Our method first represents the review with aspect-based sentiment. Then, it presents the aspect-based user similarity and product similarity. Next, the aspect-based similarities are incorporated into a matrix factorization model. To assess our proposed methods, we conducted four experiments on two datasets. The results show that the proposed model is effective and outperforms existing approaches.

**Author Contributions:** Conceptualization, methodology, funding acquisition, original draft, Q.P.; validation, formal analysis, experiment, L.Y.; resources, data curation, writing—review and editing, W.D. and X.X.; investigation, experiment, H.F. and F.Z.; experimental analysis, conclusions, K.Z.; collection of data, D.H. All authors have read and agreed to the published version of the manuscript.

**Funding:** This research was supported by Hubei Provincial Enterprise-Level Intelligent Application Excellent Young and Middle-Aged Scientific Technological Innovation Team, Natural Science Foundation of Hubei Province, No. ZRMS2019001565, the Technology Innovation Special Program of Hubei Province (No. 2022BAA044) and the Artificial Intelligence Application Research Center of Wuhan College.

**Institutional Review Board Statement:** Not applicable.

**Informed Consent Statement:** Not applicable.

**Data Availability Statement:** All the details of this work, including data and algorithm codes, are available from the corresponding author: yoyo@hubu.edu.cn.

**Acknowledgments:** The authors would like to thank the reviewers for their helpful suggestions, which have considerably improved the quality of the manuscript.

**Conflicts of Interest:** The authors declare no conflict of interest.

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
