# Peer review of "Jointly Modeling Aspect Information and Ratings for Review Rating Prediction"

_electronics, doi:10.3390/electronics11213532_

Round 1
Reviewer 1 Report
In this paper, the authors propose to transform the text of reviews into user aspects and product aspects, and combine it with sentiment analysis to generate a representation that can attain a better prediction of a review rating. I believe the authors made a good work in comparing different baselines with their method. I have some questions and comments about the work, which I believe could improve the paper.
Regarding the weight parameters alpha and beta, you show how these parameters reduce the MSE. However, in the methodology I only see that you used five-fold cross validation. Thus, the selected parameters could be biased and not represent the capabilities to generalize if it was the case that the parameters were selected on the result of the five-fold cross validation. Was there a separation into training, validation, and testing, where one part (training and validation) was used to adjust the weights parameters and then this was tested on a previously unseen part of the dataset? If not, we cannot measure if these results are good and would be able to generalize.
Regarding the presentation of the performance, is it the MSE calculated on the testing part and then averaged or is calculated from the training part. I suppose the former, but this is not clear in the paper. I would also suggest some statistical comparisons, like using a Wilcoxon test between AUPMF and UHT to determine if there is a statistically significant difference between the two methods.
Regarding the performance measure, why the use of MSE and not something that evaluates how good are the recommendations, such as Mean Average Precision, or Mean Average Recall?
What are the pros and cons against the other methods? Is it computationally faster?
Author Response
Thanks for your valuable reminder.

Reviewer 2 Report
Dear Authors,
I included some observations.
You will see them in the original message sent to the Editorial Board (below).
Thank you for your contribution.
<<
Honored Editorial Board,
As the title and the abstract suggest, this paper aims to propose a joint model that incorporates review text information with matrix factorization for review rating prediction.
After reading the entire manuscript, I think there are still some issues to consider and solve.
I will start with format considerations and then I will continue with content-related ones.
For the first class (format ones), I mention below:
-
English language and style issues - Grammarly (https://app.grammarly.com) on default settings detected only for the text block resulting from the concatenation of Title+Abstract+Keywords:
(a) 5 critical alerts (correctness issues)
(b) and 6 more advanced ones, namely:
-Unclear sentences (2),
-Word choice (2),
-Incomplete sentences (1),
-and Passive voice misuse (1).
This meant a total score of 86 (not so bad) out of 100 (maximum) for this sample above. However, since the authors do not appear to be native English speakers, I suggest a total revision of the English language and style for the entire article using Grammarly or another specialized tool;
-
The paper must follow the specific structure of the journal, namely:
Author Information, Abstract, Keywords, Introduction, Materials & Methods, Results, Discussion, Conclusions, etc., as indicated at: https://www.mdpi.com/journal/electronics/instructions -
The authors must avoid ending some sections/subsections with algorithms or figures;
-
The authors must ensure that all figures have the required resolution (minimum 1000 pixels width/height, or a resolution of 300 dpi or higher according to the Journal’s instructions: https://www.mdpi.com/journal/mathematics/instructions ); All figures in this manuscript must be revised in such terms and the authors must additionally check if such issues are not partially due to export to .pdf image settings reasons (the .pdf is the only format available for reviewers in the MDPI platform);
-
There are so many figures (7) in the paper. Those not essential for understanding the main ideas of the paper should be moved to the Appendix section. If not existing, this section must be created;
-
All references to equations/formulas must be explicitly and precisely formulated in the main text (e.g. “..see eq.N” and not “..see the eq. below:” or “.. as/as follows:”);
-
All digital object identifier (DOI) codes for all references must be explicitly specified.
For the second class of issues (content-related ones), I mention below the following:
-
The data availability statement is missing (end of the manuscript) together with full details regarding access to data (all links to all datasets used including Yelp must be explicitly included); This is very important in terms of support for replication of results;
-
I think the authors must precisely specify how the data was split into four parts (the paragraph above eq.16). I mean randomly or using various specific criteria. For robust models, both types are needed. The authors have to provide further details in the manuscript;
-
The authors must precisely specify the Operating System version (Server Standard/Essential Edition, R2? etc. - line 205);
-
The authors must specify in a short note for Figure 1 the reason for using the red font on the right (predicted instead of missing values or other reasons applicable);
-
The same for blue vs. red in Figure 2;
-
Algorithms 1 and 2 must be precisely identified in the authors’ own GitHub repository (also a replication of results reason). If not existing, the authors must create one for the entire project corresponding to this manuscript;
-
The AUPMF acronym and others existing in the manuscript must be precisely explained and at least three references to scientific papers in highly index journals must be included for each at the first occurrence in the main text;
-
A list containing all abbreviations/acronyms used must be included at the end of the manuscript;
-
The authors must use additional measurements (not just MSE) to asses the accuracy (e.g., https://doi.org/10.3390/app11052314 );
-
I think more contributions in journal papers must be cited in this research both in the Introduction and especially in the section dedicated to the interpretation of the results I think that just 32 references from which 14 in conference papers are not enough;
-
When interpreting the values obtained (especially those from sections 4.4.1 - 4.4.4) additional references to similar and previous published scientific results from papers in high-impact journals are necessary;
-
The authors' contribution should be better highlighted in the conclusions.
Thank you for the opportunity to read and check this contribution!
>>
Author Response
Thanks for your valuable reminder.

Round 2
Reviewer 2 Report
Dear Authors,
You performed some improvements.
I think the manuscript is closer to the state of being published.
I wish you all the best!